# Prior Knowledge of Diagnostic Results Does Not Bias a Subject's Interpretation of At-Home COVID-19 Antigen Tests

**Eric Lai**

Pharma-Dx, LLC, La Jolla, CA 92037, USA; elai@pharma-dx.com

**Abstract:** In developing in vitro diagnostic (IVD) tests, in particular at-home/over-the-counter (OTC) tests, one of the generally accepted exclusion criteria in IVD clinical trial protocols has been that of subjects with prior knowledge of their positive status, due to potential bias. With COVID-19 antigen tests now widely available, it is common for individuals to test themselves at home with an antigen test if they have symptoms associated with COVID-19, flu, or the common cold. These subjects would be excluded from participation in COVID-19-related clinical trials (i.e., COVID-19 alone or any multiplex tests that include COVID-19). This study compiled the results of four clinical trials to assesses whether potential bias (positive or negative) exists in at-home antigen COVID-19 testing if someone has a prior diagnostic COVID-19 result. The results clearly demonstrated that knowledge of prior COVID-19 diagnostic results did not affect the accuracy of the test result interpretations nor the clinical performance of the at-home antigen test products. This is an important finding that supports the inclusion of these subjects in future COVID-19 diagnostic clinical trials and the FDA's recommendation of serial antigen testing to improve antigen test performance.

**Keywords:** COVID-19; SARS-CoV-2; enrollment bias; at-home antigen tests; serial testing





## 1. Introduction

Since the outbreak of the novel coronavirus SARS-CoV-2 in 2019, the United States Food and Drug Administration (FDA) has authorized 44 COVID-19 at-home/over-the-counter (OTC) antigen tests as of 3 February 2023 [1]. On 25 October 2021, the Biden–Harris Administration and the U.S. Department of Health and Human Services (HHS) announced the creation of the National Institutes of Health (NIH) Rapid Acceleration of Diagnostics (RADx®) Independent Test Assessment Program (ITAP) [2] to accelerate regulatory review and availability of high-quality, accurate, and reliable at-home antigen tests for COVID-19 nationwide. Concurrently, on 15 December 2021, the Administration announced it would provide online ordering and distribution of 500 million free at-home rapid tests for Americans, effective 19 January 2022 [3]. Since the first OTC antigen test emergency use authorization (EUA) on 24 December 2021 [4], ITAP has assisted high volume antigen test manufacturers to receive 12 EUAs, and these companies have increased capacity by more than 3.2 billion tests to the U.S. as of December 2022 [5]. Subsequent free release of an additional 1 billion tests to the U.S. public have changed the COVID-19 testing landscape, and COVID-19 testing is no longer a bottleneck in the U.S. [6].

In developing in vitro diagnostic (IVD) tests, in particular at-home/OTC tests, one of the generally accepted exclusion criteria in IVD clinical trial protocols has been that of subjects with prior knowledge of their positive status. The rationale behind this practice is the general belief that a subject might be biased when reading their antigen test if that individual knew whether they were positive or negative with the previous test. With antigen tests now widely available, it is common for individuals to test themselves with a COVID-19 antigen test at home if they have symptoms associated with COVID-19, flu, or the common cold. These subjects would be excluded from participation in COVID-19-related diagnostic clinical trials (i.e., COVID-19 alone or any multiplex tests that include

COVID-19). This potentially creates two unwanted consequences: (1) the potential pool of eligible subjects for diagnostic clinical trial enrollment is reduced (i.e., higher enrollment numbers to meet eligible subject minimum, increased cost, and longer trial time) and (2) the trials do not represent the real-world situation. More importantly, recent FDA guidance on at-home antigen testing recommends repeat/serial testing within two to three days to minimize false negative results and to improve positive performance of the tests [7]. A potential bias from prior testing, if it exists, would question the validity of the FDA serial testing recommendation. As this situation is a recent phenomenon due to the widespread availability of at-home antigen tests, there are no data to support or disprove potential bias. Therefore, it is important to assess whether potential bias (positive or negative) exists in at-home antigen COVID-19 testing if someone has a prior diagnostic COVID-19 result.

## 2. Materials and Methods

Enrollment and clinical performance of four different COVID-19 at-home antigen tests were evaluated in separate diagnostic clinical trials performed by three contract research organizations (CROs) at 21 clinical sites in 18 cities. The trials were performed using the ITAP COVID-19 clinical trial protocol Version 11 in an "all comers" design and funded by the NIH ITAP program. The trials were approved by Institutional Review Boards and all participants provided written informed consent. All trials used the Roche cobas® SARS-CoV-2 & Influenza A/B combo assay as the comparator. The comparator testing was performed at clinical testing laboratories and the instruments were calibrated and cycle threshold (Ct) standardized with a well-defined 10 serial dilutions panel and testing of each dilution with 20 replicates. The low positive cutoff at Ct $\geq$ 30 was analyzed and defined in collaboration with the FDA. These four diagnostics trials were designed in consultation with the FDA and were the only COVID-19 diagnostics trials with enrollment of subjects with diagnostic results obtained in the three days prior to enrollment out of more than 30 NIH-sponsored COVID-19 diagnostics clinical trials. After subject consent and enrollment, subjects were asked whether they had any COVID-19 test results (e.g., PCR, LFA, etc.) in the prior three days and whether the results were positive or negative. The type of prior COVID-19 test(s) was not recorded because most subjects could not accurately provide the information. The results from the investigational at-home antigen test device were reported and recorded by the subjects in an at-home setting. A healthcare provider (HCP) then immediately examined the device, recorded their interpretation of the test results, and took a picture of the device. At the end of the trial, the pictures were examined by a second independent individual (or a regulatory agency if submitted for authorization) to assess the accuracy of the subjects' reported results of the investigational device. The trial results have not been published but the data from these trials were used to support successful Emergency Use Authorizations.

## 3. Results

A total of 1327 subjects were enrolled in the diagnostic trials (Table S1 in Supplementary Materials). The demographic data received for this analysis are shown in Table 1. Only sex and age were available from all four trials. Other demographic data such as education level, income, and race were not available in one or more of the trials. The 95% confidence intervals overlap across all groups, suggesting there were no significant differences in the sex and age of these groups. Of the 1327 subjects, 1312 subjects reported whether they had a diagnostic test in the prior three days. Of those 1312 subjects, 124 subjects (9.5%) reported they had known COVID-19 diagnostic results within the prior three days; 45 subjects (36.3%) had known positive results, and 79 subjects (63.7%) had known negative results. Examination of the photos of the 124 subjects at the end of the trials by an independent reviewer resulted in one discrepancy between the subject/HCP's interpretations and the reviewer's.

**Table 1.** Sex and age of the subjects in the dataset.

| | Complete Dataset | Subjects without Prior Known COVID-19 Results | Subjects with Prior Known COVID-19 Results | Subjects with Known COVID-19 Positive Results | Subjects with Known COVID-19 Negative Results |
|---|---|---|---|---|---|
| Sex | | | | | |
| Female | 786 (60.4%) | 697 (59.8%) | 82 (67.8%) | 25 (59.5%) | 57 (72.2%) |
| Male | 515 (40.6%) | 468 (40.2%) | 39 (32.2%) | 17 (40.5%) | 22 (27.8%) |
| Total = | 1301 [1] | 1165 [2] | 121 [3] | 42 [3] | 79 |
| Age | | | | | |
| Average (SD) | 36.2 (19.0) | 36.3 (19.4) | 35.3 (16.6) | 32.3 (14.9) | 41 (18.3) |
| Min, Max | 2, 89 | 2, 89 | 10, 79 | 10, 72 | 10, 79 |

[1] 26 subjects did not report their sex. [2] 15 subjects did not report whether they had a known result in the prior three days. [3] Three subjects with known prior diagnostics results did not report their sex.

No bias was observed with subjects with known COVID-19 diagnostic results in the prior three days.

The result of the investigational antigen test devices of the 45 subjects with known COVID-19 positive diagnostic results obtained in the three days prior to enrollment in the diagnostic clinical trials is shown in Table 2.

**Table 2.** Analysis of the 45 subjects with prior known COVID-19 positive diagnostic results [1].

| Investigational Antigen Device Results [1] | Comparator Results | Concordance |
|---|---|---|
| 22 positives | 21 positives | True positives |
| | 1 negative | False positive |
| 21 negatives | 16 negatives | True negatives |
| | 5 positives (Ct = 27.6, 32.1, 33.5, 35.2, and 35.3) | False negatives |

[1] Two subjects had invalid results (absence of the control line as defined by the device's Instructions For Use) and were not included in the analysis.

Of the 43 subjects with known COVID-19 positive diagnostic results obtained in the three days prior to enrollment in the diagnostic clinical trials and with valid investigational and comparator results, 22 subjects (51.2%) reported investigational antigen device positive results while 21 subjects (48.8%) reported device negative results. More importantly, 21 of the 22 subjects who reported investigational antigen device positive results were true positives when compared to the comparator while one result was a false positive (i.e., the comparator result was negative). Of the 21 subjects who reported investigational antigen device negative, five results were false negatives (i.e., the comparator results were positive). However, four of these false negative subjects had comparator Ct values greater than 32 (the comparator platform was calibrated with low positive samples at Ct ≥ 30) and were not expected to be detected by the antigen tests. Thus, only one false negative should be considered to be a false negative.

The results of the investigational antigen test devices of the 79 subjects with known COVID-19 negative diagnostic results obtained in the three days prior to enrollment in the diagnostic clinical trials are shown in Table 3.

Of the 79 subjects with known COVID-19 negative diagnostic results obtained in the three days prior to enrollment in the diagnostic clinical trials and with valid investigational and comparator results, 77 subjects reported investigational antigen test device negative results while 2 subjects reported device positive results. Of the 77 subjects who reported investigational antigen device negative, four results were false negatives (i.e., the comparator results were positive). However, all of these false negative subjects had comparator Ct values of greater than or equal to 30 (the comparator platform was calibrated with low

positive samples at Ct $\geq$ 30) and were not expected to be detected by the antigen tests. More importantly, of the two subjects who reported investigational antigen device positive, one was a true positive and the other was a true negative when compared to the comparator. An equal number of subjects who reported positive test results reported true positive (one subject) and negative results (one subject).

**Table 3.** Analysis of the 79 subjects with prior known COVID-19 negative results.

| Investigational Antigen Device Results | Comparator Results | Concordance |
|---|---|---|
| 77 Negatives | 73 Negatives | True negatives |
| | 4 Positives (Ct = 30.0, 32.0, 33.0, and 36.0) | False negatives |
| 2 Positives | 1 Positive | True Positive |
| | 1 Negative | False Positive |

*Clinical Performance*

Since the most important assessment of a clinical trial is the clinical performance of the investigational device, it is important to assess whether subjects with prior known COVID-19 diagnostic results might have different clinical performance results as compared to subjects without prior known results. To assess this possibility, the clinical performance (e.g., positive percent agreement, negative percent agreement, etc.) was calculated in the two subgroups and is shown in Table 4.

**Table 4.** Clinical performance of the subgroups.

| | Whole Dataset (*n* = 1327); Evaluable Dataset (*n* = 1294) [1] | | Subjects with Prior Known COVID-19 Diagnostic Results (*n* = 124); Evaluable Dataset (*n* = 122) [2] | | Subjects without Prior Known COVID-19 Diagnostic Results (*n* = 1188); Evaluable Dataset (*n* = 1172) [3] | |
|---|---|---|---|---|---|---|
| | Comparator + | Comparator − | Comparator + | Comparator − | Comparator + | Comparator − |
| Device + | 169 | 10 | 22 | 2 | 147 | 8 |
| Device − | 59 | 1056 | 9 | 89 | 50 | 967 |
| | 228 | 1066 | 31 | 91 | 197 | 975 |
| PPA [4] | 74.1% | | 71.0% | | 74.6% | |
| PPA (95% CI) [5] | 68.1–80.1% | | 53.4–88.6% | | 68.1–81.1% | |
| NPA [6] | 99.1% | | 97.8% | | 99.2% | |
| Low Positive (Ct $\geq$ 30) | 56 | | 9 | | 47 | |
| % of low positives | 24.5% | | 29.0% | | 23.7% | |

[1] 33 subjects had invalid or no device results; [2] 2 subjects had invalid device results; [3] 15 subjects did not provide information on whether they had prior diagnostic results; [4] positive percent agreement; [5] 95% confidence interval; [6] negative percent agreement.

The results in Table 4 show that there were no differences in clinical performance within the entire dataset and among the subgroups with or without prior COVID-19 diagnostic results. If there was bias among the subjects with prior known COVID-19 diagnostic results, one would expect better clinical performance in that group when compared to the whole dataset or subjects without prior known COVID-19 diagnostic results.

## 4. Discussion

There exists a long-held general perception that it is not acceptable or desirable for subjects to have knowledge of their diagnostic results prior to enrollment in IVD clinical trials. However, with the widespread availability of free at-home antigen tests since January 2022, it is neither cost efficient nor realistic to exclude subjects with prior knowledge of their COVID-19 test results. For example, the CRO would need to enroll approximately 10% more subjects in the four diagnostic trials described in this study to achieve study objectives with this exclusion criterion. Furthermore, the clinical trial design and testing setting would not be representative of the real-world environment where a significant percentage of the public might have tested themselves at home before seeking medical care.

This study examined whether the knowledge of a recent (i.e., within three days) diagnostic COVID-19 result could potentially influence a subject's interpretation of an at-home antigen test. Of the 43 subjects with known COVID-19 positive diagnostic results and with valid investigational and comparator results, an approximately equal number of subjects reported positive (22 subjects) and negative (21 subjects) investigational antigen device results. This suggests that prior knowledge of a known COVID-19 positive test result did not influence the subject's interpretation of the antigen test. Similarly, 77 of the 79 subjects with known COVID-19 negative diagnostic results reported negative results with the investigational antigen device. These results demonstrated that there was no positive or negative bias in the subject's interpretation of the at-home antigen test with known COVID-19 diagnostic results obtained in the past three days.

Implications of this study include understanding whether bias associated with known prior COVID-19 test results could affect the interpretation and clinical performance of the FDA's recent recommendation to use serial antigen testing within 24–48 h of the prior test result. In collaboration with the FDA, four of the recent NIH RADx COVID-19 ITAP clinical trials were designed to include subjects with prior known COVID-19 diagnostic results. These diagnostic trials included multiple steps of reading and recording antigen test results by the subjects and two independent observers. The results clearly demonstrated that knowledge of COVID-19 diagnostic results obtained in the past three days did not affect the accuracy of the test result interpretations nor the clinical performance of the at-home antigen test products. This is an important finding that supports the inclusion of these subjects in future diagnostic clinical trials and the FDA's recommendation of serial antigen testing to improve antigen test performance.

**Supplementary Materials:** The following supporting information can be downloaded at: https://www. mdpi.com/article/10.3390/covid3040045/s1, Table S1: COVID-19 Antigen Diagnostic Test Results.

**Funding:** The clinical trials included in this study were supported by the National Institute of Biomedical Imaging and Bioengineering of the National Institute of Health as part of the ITAP program to speed the authorization and commercialization of at home COVID-19 antigen tests (Grant contracts #75N92022C00027 for the author and #75N92022D00010 for the clinical trials). The funders had no role in the decision to submit the work for publication, and the views expressed herein are the author's and do not necessarily represent the views of the National Institutes of Health or the US Department of Health and Human Services.

**Institutional Review Board Statement:** The clinical trials were performed by the manufacturers of the four at-home antigen tests. The trials were conducted in accordance with the Declaration of Helsinki, reviewed and approved by Institutional Review Boards. The trial results were submitted to the US FDA for Emergency Use Authorization Approvals. The manufacturers of the four at-home antigen tests have agreed to have their clinical trial data included in this study.

**Informed Consent Statement:** Informed consent was obtained from all subjects involved in the study.

**Data Availability Statement:** The original clinical trial data from the manufacturers have been submitted to the FDA for Emergency Use Authorization Approvals. The de-identified data described in this publication will be made available if requested.

**Acknowledgments:** The author gratefully acknowledges NIH colleagues Bruce Tromberg, Jill Heemskerk, Todd Merchak, Michael Wolfson, Bill Heetderks, and FDA colleagues Timothy Stenzel, Kristian Roth, Silke Schlottmann and Brittany Goldberg for their support of this study. The author is grateful for the input and review of the manuscript provided by D'lynne Plummer, Julie Wilkinson, Emily Kennedy, and Christine Cooper.

**Conflicts of Interest:** The author is a consultant to the NIH RADx® Tech and ITAP programs and is the clinical lead of the NIH ITAP clinical trial studies. The author declares that there is no conflict of interest.

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
