# Peer review of "Prior Knowledge of Diagnostic Results Does Not Bias A Subject’s Interpretation of At-Home COVID-19 Antigen Tests"

_covid, doi:10.3390/covid3040045_

Round 1
Reviewer 1 Report
I have read the article "Prior Knowledge of diagnostic results does not bias a subject's interpretation of at-Home COVID-19 antigen test". The work had great interest, and the introduction shows interest in carrying out this study. Anyway, there are some points that I would like to be clarified:
1.-lines 75-76: "The type of prior COVID-19 test(s) was not recorded". The author should discuss whether this fact may be important in the final result.
2.-lines 82-83: why were the results of these trials not published? How can anyone access this data to check the results?
3.-Some of the statements in the results section are interpretations of the results. Regardless of whether the interpretation is correct, I think these statements should be in the discussion section. An example of these interpretations are:
Lines 109-111: "This suggests that prior..."
Lines 120-121: "These results demonstrated..."
lines 136-137: "This suggests that prior..."
Lines 139-140: "These results demonstrate ..."
Lines 142-144: " The results shown above clearly demonstrate..."
4.- Table 2: two subjects had invalid results. What was the reason for these invalid results?
Author Response
1.-lines 75-76: "The type of prior COVID-19 test(s) was not recorded". The author should discuss whether this fact may be important in the final result.
The clinical trial protocols were developed in consultation with the FDA and the type of prior COVID-19 test(s) used by a subject was never recorded in clinical trials since the beginning of COVID-19. Thus there is no data to discuss regarding this topic.
2.-lines 82-83: why were the results of these trials not published? How can anyone access this data to check the results?
The data supporting this publication came from 4 clinical trials that have submitted to the FDA and have received EUA authorization. Whether the clinical trials will be published is the decision of the companies. The data in this paper have been presented to the FDA on Jan 31, 2023 and a copy of the paper has been submitted to the FDA for review on Feb 12, 2023. A deidentified version of the data without personal health information could be submitted as supplemental material if needed.
3.-Some of the statements in the results section are interpretations of the results. Regardless of whether the interpretation is correct, I think these statements should be in the discussion section. An example of these interpretations are:
Lines 109-111: "This suggests that prior..."
Lines 120-121: "These results demonstrated..."
lines 136-137: "This suggests that prior..."
Lines 139-140: "These results demonstrate ..."
Lines 142-144: " The results shown above clearly demonstrate..."
The author agrees with the reviewer’s comment and has rewritten the Results and Discussion sections.
4.- Table 2: two subjects had invalid results. What was the reason for these invalid results?
The results were invalid due to the absence of the control line as defined by the investigational device’s Instruction For Use).
Reviewer 2 Report
When developing in vitro diagnostic tests , and using exclusion criteria that are accepted in clinical protocols . Unfamiliar positivity and negativity criteria can trigger a process change in the assay . In particular if the procedure is done in house without the aid of specific kits. In this context it is very important to have the criteria that will be considered positive or negative. With COVID-19 antigen tests now widely available, it is common for individuals to test themselves and at home with an antigen test if they have symptoms associated with COVID-19, influenza, or the common cold. The aforementioned population has been excluded in Covid-related clinical trials of any kind .This study compiled the results of 4 clinical trials to assess whether there is potential bias (positive or negative) when testing at home . What was interesting about the study is that there is not a difference . The work clearly shows that previous results influence interpretation. The quality of the previous tests are relevant even if done at home clearly showing the necessity of the tests used. The inclusion of the topic is relevant as well as the improvement of rapid tests for used diagnosis .
Suggested title:
Prior knowledge of diagnostic results does not interfere the interpretation of at-home COVID-19 antigen tests
Author Response
The reviewer’s suggested title change to “Prior knowledge of diagnostic results does not interfere the interpretation of at-home COVID-19 antigen tests” was discussed with the FDA. The word “Bias” in the original title is an important regulatory term that the FDA would like to keep. Proposed title change: “Prior knowledge of diagnostic results does not bias the interpretation of at-home COVID-19 antigen tests”.
Round 2
Reviewer 1 Report
The author has reasonably answered to the questions raised in the review. I think the article can be published in its current form.